# Humoral Immunity in Immunosuppressed IBD Patients after the Third SARS-CoV-2 Vaccination: A Comparison with Healthy Control Subjects

**DOI:** 10.3390/vaccines11091411

**Published:** 2023-08-24

**Authors:** Richard Vollenberg, Eva Ulla Lorentzen, Joachim Kühn, Tobias Max Nowacki, Jörn Arne Meier, Jonel Trebicka, Phil-Robin Tepasse

**Affiliations:** 1Department of Medicine B for Gastroenterology, Hepatology, Endocrinology and Clincial Infectiology, University Hospital Muenster, 48149 Muenster, Germany; jonel.trebicka@ukmuenster.de (J.T.); phil-robin.tepasse@ukmuenster.de (P.-R.T.); 2Institute of Virology, University Hospital Muenster, 48149 Muenster, Germanyjoachim.kuehn@ukmuenster.de (J.K.); 3Department of Medicine, Gastroenterology, Marienhospital Steinfurt, 48565 Steinfurt, Germany

**Keywords:** SARS-CoV-2, COVID-19, vaccination, IBD patients, seroconversion, humoral immune response

## Abstract

Introduction: The COVID-19 pandemic is a result of severe acute respiratory syndrome coronavirus 2 (SARS-CoV-2). Vaccination against COVID-19 is crucial for preventing severe illness and controlling the pandemic. This study aimed to examine how immunosuppressed patients with inflammatory bowel disease (IBD) responded to the third mRNA vaccination against SARS-CoV-2. The patients were undergoing treatments such as anti-TNF (infliximab, adalimumab), anti-α4ß7 integrin (vedolizumab), anti-IL12/23 (ustekinumab) and azathioprine (purine analog). Their responses were compared to those of healthy individuals. Methods: In this prospective study, 81 IBD patients and 15 healthy controls were enrolled 2–4 months after receiving the third mRNA vaccination. This study measured IgG antibody levels against the SARS-CoV-2 spike protein’s receptor binding domain (RBD) and assessed potential neutralization capacity using a surrogate virus neutralization test (sVNT). Results: Overall, immunosuppressed IBD patients (without SARS-CoV-2 infection) exhibited significantly lower levels of anti-S-IgG (anti-RBD-IgG) and binding inhibition in the sVNT after the third vaccination compared to healthy controls. Patients under anti-TNF therapy showed notably reduced anti-S-IgG levels after the booster vaccination, in contrast to those receiving ustekinumab and azathioprine (*p* = 0.030, *p* = 0.031). IBD patients on anti-TNF therapy demonstrated significantly increased anti-S-IgG levels following prior SARS-CoV-2 infection (*p* = 0.020). Conclusion: Even after the third vaccination, immunosuppressed IBD patients exhibited diminished humoral immunity compared to healthy controls, especially those on anti-TNF therapy. Cases of penetrating infections led to considerably higher antibody levels in IBD patients under anti-TNF therapy compared to uninfected patients. Further investigation through prospective studies in immunosuppressed IBD patients is needed to determine whether this effectively safeguards against future infections or severe disease.

## 1. Introduction

The COVID-19 pandemic was caused by the SARS coronavirus-2 [1]. A large proportion of people become mildly ill, but some patients develop a severe course of illness after infection. This can include respiratory failure, hyperinflammatory syndrome leading to multiorgan failure and death [2,3,4]. Antiviral agents such as Paxlovid, remdesivir and monoclonal antibodies (mabs) such as sotrovimab have shown a mitigating effect on disease progression in the early phase of infection. They can partly reduce mortality; however, immune escape mutants of SARS-CoV-2 have increasingly become resistant to neutralization by mabs [5,6]. Anti-inflammatory drugs such as dexamethasone and tocilizumab may reduce mortality in the late, proinflammatory phase of disease [7,8]. However, vaccination is the most effective measure to prevent severe courses. In particular, the mRNA vaccines mRNA-1273 and BNT162b2 have shown high efficacy [9,10,11,12]. Full vaccine protection is achieved with these vaccines after two applications. However, after a latency of several months, a decrease in vaccine protection is shown, so that a booster vaccination is necessary after two basic immunizations [13]. Patients receiving therapy with immunosuppressive drugs have frequently been shown to have reduced humoral immune responses to the above-mentioned vaccines. Patients with inflammatory bowel disease (Crohn’s disease and ulcerative colitis) require immunosuppressive therapies for disease control. This leads to significant uncertainty among healthcare providers regarding the use and selection of immunosuppressive therapies in IBD patients. In two previous studies, we were able to demonstrate that notably, IBD patients receiving anti-TNF-substance therapy exhibit impaired humoral immunity after dual vaccination [13,14,15]. Existing studies on IBD patients after a third vaccination display a heterogeneous data landscape on this matter. For the majority of IBD patients, the COVID-19 booster vaccination occurred over a month ago. However, studies with extended postvaccination follow-up periods are currently lacking. Both healthcare providers and patients are grappling with substantial uncertainty concerning the sustained humoral immunity after COVID-19 booster vaccination in immunosuppressed IBD patients.

The objective of this study was to compare the humoral immunity in IBD patients 60–120 days after the COVID-19 booster vaccination based on their existing immunosuppressive therapy with healthy control subjects. Furthermore, the impact of a prior SARS-CoV-2 infection on humoral immunity after the third vaccination was examined in immunosuppressed patients.

## 2. Methods

### 2.1. Study Subjects and Samples

Serum samples in this monocentric, prospective study were collected 60 to 120 days after a third vaccination against SARS-CoV-2 from IBD patients in the IBD outpatient clinic of the Department of Gastroenterology, Hepatology, Endocrinology and Clinical Infectiology at Münster University Hospital (n = 81) and from healthy control subjects (n = 15). Vaccination of patients and control subjects was performed independently of this study after consultation with the respective treating physician. All study participants were vaccinated exclusively with the mRNA vaccines mRNA-1273 and BNT162b2 and were questioned about previous SARS-CoV-2 infection. In addition, sera from all participants were tested for the presence of IgG antibodies to the nucleocapsid antigen (anti-N IgG) of SARS-CoV-2. Previous SARS-CoV-2 infection (defined as a positive nasopharyngeal swab SARS-CoV-2 PCR test) was detected in n = 12 patients and n = 3 healthy controls. No IBD patients or healthy controls currently infected with COVID-19 were included. All IBD patients and healthy controls were queried for serious vaccine adverse events. Serious vaccine adverse events were defined as those that are fatal or life-threatening, require hospitalization or lead to permanent damage. Only patients receiving therapy with anti-TNF agents (adalimumab, infliximab), anti-α4ß7 integrin (vedolizumab), anti-IL12/23 (ustekinumab) or azathioprine (purine analog) were included (n = 5 patients receiving other therapy were excluded). In all patients, therapy was started at least 12 weeks before the first vaccination and was not changed thereafter. Finally, n = 60 patients and n = 12 healthy controls without prior SARS-CoV-2 infection and 12 patients/3 controls after COVID-19 infection were included (Figure 1). Crohn’s disease patients with a Crohn’s disease activity index (CDAI) ≥ 150 were defined as having active Crohn’s disease, and ulcerative colitis patients with a Mayo score ≥ 1 were defined as having active ulcerative colitis. This study was approved by the local ethics committee (Münster University Hospital: 2021-039-f-S), and study participants gave written informed consent.

### 2.2. Quantification of Serum Markers

To identify patients with prior SARS-CoV-2 infection, IgG antibodies to the nucleocapsid antigen (anti-N IgG) were qualitatively determined in all sera by a commercial, CE/IVD certified chemiluminescence microparticle assay (CMIA) (SARS-CoV-2 IgG assay, Abbott Diagnostics, Abbott Park, North Chicago, IL, USA). To assess the humoral response to SARS-CoV-2 vaccination and/or infection, IgG antibodies against the SARS-CoV-2 receptor binding domain (RBD) of the spike protein subunit S1 were quantified by CMIA (SARS-CoV-2 IgG II Quant assay, Abbott Diagnostics). The assays were performed according to the manufacturer’s manual on an Architect device (Abbott, Chicago, IL, USA) as previously described [13,14]. 

The CE/IVD certified cPass^TM^ SARS-CoV-2 Neutralization Antibody Detection Kit (GenScript Biotech, Leiden, The Netherlands) was applied to characterize the sera with respect to their capacity to block the interaction between the RBD and the human host cell receptor protein ACE2 and, thus, infection. The cPass assay determines ACE2-RBD binding inhibition in an enzyme-linked immunosorbent assay (ELISA) format, can be conducted under routine biosafety level 2 conditions and correlates well with conventional and pseudo-virus-based virus neutralization assays [16,17,18,19]. The cut-off value is set at 30% inhibition for the CE/IVD version of the assay by its manufacturer, GenScript.

### 2.3. Statistical Examination

Categorical variables were compared using the chi-square test or Fisher’s exact test (expressed as absolute numbers and percentages). Continuous variables were assessed using the t-test (presented as medians with interquartile ranges (IQR)) when normally distributed and the Mann–Whitney U test (Wilcoxon) when non-normally distributed. For comparisons involving more than two groups, the Kruskal–Wallis test was employed. Subgroup comparisons were subjected to the Bonferroni correction post hoc test in instances of equal variance (determined by Levene’s test), while the Games–Howell test was applied when variance was unequal. Point biserial correlation was used for non-normally distributed, nominal/interval scaled variables; biserial correlation for nominally/ordinally distributed variables; and the phi coefficient for nominally scaled variables. All tests were two-sided, with a *p*-value < 0.05 signifying statistical significance. Statistical analyses were conducted using SPSS 26 (IBM, Chicago, IL, USA).

## 3. Results

### 3.1. Cohort Characteristics

Between 1 January 2021 and 10 October 2022, 81 IBD patients and 15 healthy controls were enrolled in this study at the time of visit to Münster University Hospital. 

Included in this analysis were 60 participants without prior SARS-CoV-2 infection who were treated with anti-TNF (n = 37; of these, adalimumab n = 12, infliximab n = 25), vedolizumab (n = 11), azathioprine (n = 3) or ustekinumab (n = 9) and 12 healthy controls. Previous SARS-CoV-2 infection was detected in 12 IBD patients (anti-TNF n = 9; of these, adalimumab n = 1, infliximab n = 8, vedolizumab n = 2, ustekinumab n = 1) and 3 healthy controls. An antibody test was performed between 60 and 120 days after the third dose of SARS-CoV-2 vaccine. 

The mean age of the included IBD patients without previous SARS-CoV-2 infection was 46 years (healthy controls 50 years, *p* = 0.678); 37 (62%) of the 60 IBD patients were male (controls n = 5) and 23 (48%) were female (controls n = 7). There were no significant differences between the groups in terms of sex distribution and body mass index (BMI). Of the patients, 37 (62%) had Crohn’s disease (5% with active disease) and 23 (38%) had ulcerative colitis (48% with active disease). The healthy control subjects had no chronic diseases and were not taking any medications. Among the IBD patients, cardiovascular disease was the most common comorbidity (17%) (Table 1). Oral prednisolone therapy was present in 4 (6%), oral/local budesonide therapy in 1/3 (1/4%) and oral/local mesalazine therapy in 25/5 (34/7%) of IBD patients. Regardless of the existing immunosuppressive therapy, there were no significant differences in IBD patients regarding the existing comedication (prednisolone, budesonide and mesalazine, Table 1).

Of the 12 patients with prior SARS-CoV-2 infection, 9 were on anti-TNF therapy (of these, adalimumab n = 1, infliximab n = 8), and 3 had other therapies (2 vedolizumab, ustekinumab). Crohn’s disease was present in 83% of patients (no patient with active disease), and ulcerative colitis was present in 17% (no patient with active disease). Compared to healthy controls after previous SARS-CoV-2 infection, there were no significant differences regarding sex, age and BMI (Table 1).

### 3.2. Humoral Immunity in Immunosuppressed IBD Patients after SARS-CoV-2 Vaccination: Comparison with Healthy Controls

In the observed period after the third dose of an mRNA SARS-CoV-2 vaccine, SARS-CoV-2 S-IgG (AU/mL) and binding inhibition (percent) by sVNT were significantly decreased in IBD patients without previous SARS-CoV-2 infection compared with healthy controls (SARS-CoV-2 anti-S-IgG: IBD patients median 5956 AU/mL (IQR 1996–12461 AU/mL) vs. controls median 9627 AU/mL (IQR 7420–25950 AU/mL); *p* = 0.034; sVNT: IBD patients median 96% (94–97%) vs. controls 97% (IQR 96–97%); *p* = 0,012; Table 2, Figure 2). The anti-S-IgG seroconversion rate was very high in all IBD subgroups and in healthy controls and did not differ significantly in either group (seroconversion rate anti-S-IgG: IBD patients 98% vs. controls 100%; *p* = 1.000; Table 2). 

Comparison of healthy controls with each IBD treatment group (anti-TNF, vedolizumab, ustekinumab and azathioprine) showed no significant differences in anti-S-IgG levels and binding inhibition by sVNT. In trend, IBD patients under therapy with vedolizumab and ustekinumab showed higher to equal anti-S-IgG levels and binding inhibition compared to controls, and under therapy with azathioprine and anti-TNF showed reduced anti-S-IgG titers and binding inhibition (anti-S-IgG: controls median 9627 AU/mL (IQR 7420–25,950) vs. vedolizumab 11,388 AU/mL (7319–22,027 AU/mL) vs. anti-TNF 3410 AU/mL (1843–6617 AU/mL) vs. azathioprine 5903 AU/mL (989–5903 AU/mL) vs. ustekinumab 20,928 AU/mL (7882–23,558 AU/mL); sVNT: controls median 97% (IQR 96–97%) vs. vedolizumab 96% (96–97%) vs. anti-TNF 95% (90–96%) vs. azathioprine 96% (83–96%) vs. ustekinumab 97% (96–97%)). IBD patients on anti-TNF therapy (median 3410 AU/mL (IQR 1843–6617 AU/mL); *p* = 0.030) or azathioprine (5903 AU/mL (989–5903 AU/mL); *p* = 0.031) showed significantly decreased anti-S-IgG levels compared with patients on therapy with ustekinumab (20,928 AU/mL (7882–23,558 AU/mL)). There were no significant differences between groups regarding binding inhibition by sVNT. 

Regarding the existing anti-TNF therapy (adalimumab versus infliximab), no significant differences were observed among IBD patients without prior COVID-19 infection in terms of anti-S-IgG (*p* = 1.0) and sVNT levels (*p* = 0.810) (refer to Appendix A).

The RBD-ACE2 binding inhibition, anti-S-IgG levels and seroconversion rates (sVNT, S-IgG) of IBD patients and healthy controls after SARS-CoV-2 infection did not differ significantly (Table 2). 

Healthy controls without prior SARS-CoV-2 infection showed significantly higher binding inhibition than controls after prior infection (97% (96–97%) vs. 96% (95–96%); *p* = 0.036) and tended to have lower anti-S-IgG levels (9627 AU/mL (7420–25,950 AU/mL) vs. 29,365 AU/mL (1934–29,365 AU/mL); *p* = 0.840). Comparing all IBD patients, there was a trend toward higher anti-S-IgG titers in patients after previous SARS-CoV-2 infection (8409 AU/mL (5314–12,846 AU/mL) vs. 5956 AU/mL (1996–12,461 AU/mL)). However, this difference was not statistically significant (*p* = 0.280; Figure 3). IBD patients on anti-TNF therapy showed significantly higher anti-S-IgG levels after previous SARS-CoV-2 infection (3410 AU/mL (1843–6617 AU/mL)) compared to patients on anti-TNF therapy without previous infection (8434 AU/mL (6227–20,155 AU/mL); *p* = 0.020) (Appendix A). Regarding binding inhibition in the sVNT, there were no significant differences in the two IBD groups.

## 4. Discussion

Several studies have demonstrated that patients with inflammatory bowel disease (IBD) undergoing anti-TNF therapy; combination therapy of anti-TNF and thiopurines; or tofacitinib exhibit a significantly impaired humoral immune response following the second SARS-CoV-2 vaccination [13,14,20,21]. Limited and partly conflicting data exist regarding the humoral immune response in immunosuppressed IBD patients after a third SARS-CoV-2 vaccination. However, previous studies have only investigated median follow-up times of 30–40 days after the third vaccination. Given that the third vaccination occurred further in the past for the majority of IBD patients, uncertainty persists regarding the existing humoral protective effect in immunosuppressed patients. Schell et al. (HERCULES study) demonstrated that in patients with IBD, median anti-S-IgG levels were elevated 37 days after a third mRNA vaccination (homologous prime-boost regimen) compared to those who received only two vaccinations. Notably, patients treated with steroids, anti-TNF agents or a combination of anti-TNF substances also exhibited reduced anti-S-IgG titers after the third vaccination [22]. Long et al. showed a significant increase in anti-S-IgG levels in 408 patients with a median of 48 days after a third vaccination (heterologous regimen with mRNA and vector vaccines) compared to their status after only two vaccinations. Consistent with earlier findings, patients treated with anti-TNF agents also had lower anti-S-IgG levels in this context [23]. In contrast, a recent Canadian study observed no significant reduction in anti-SARS-CoV-2 spike antibody levels after three doses in IBD patients receiving anti-TNF therapy [24]. Alexander et al. demonstrated a significant increase in antibody levels in 352 IBD patients under immunosuppressive therapy 28–49 days after a third SARS-CoV-2 vaccination compared to patients after the second vaccination. Patients under anti-TNF therapy exhibited lower antibody levels compared to a healthy control group [25]. Kennedy et al. observed a significant increase in anti-S-IgG levels five weeks after a third mRNA vaccination (homologous regimen) in a cohort of 918 IBD patients receiving anti-TNF therapy and 442 patients receiving vedolizumab therapy compared to values after the second vaccination. Particularly, patients under anti-TNF therapy exhibited significantly lower values compared to those under vedolizumab therapy and also had a higher incidence of breakthrough infections [26]. Similarly, Liu et al. (VIP study) found that among 871 patients receiving anti-TNF therapy and 417 patients receiving vedolizumab therapy, there were significant reductions in anti-S-IgG levels and increased breakthrough infections after a third vaccination when anti-TNF agents were used [27]. In summary, the majority of studies reported an increase in anti-S-IgG titers in IBD patients after a median of 40 days following the third vaccination. In most studies, patients under anti-TNF therapy exhibited reduced anti-S-IgG titers.

In our study, we investigated an IBD cohort during the period of 60 to 120 days after the third vaccination, which extends beyond the timeframes examined in the aforementioned studies. This temporal differentiation offers novel insights into the longer-term course of humoral immunity following the third COVID-19 vaccination. Our study revealed that the group of immunosuppressed IBD patients (treated with vedolizumab, ustekinumab, anti-TNF or azathioprine) and without a previous COVID-19 infection exhibited significantly lower mean anti-S-IgG levels compared to healthy control individuals after the third SARS-CoV-2 mRNA vaccination. In the subgroup analysis of IBD patients based on their existing immunosuppressive therapy, patients treated with infliximab or azathioprine had significantly lower antibody levels (anti-S-IgG) against wild-type SARS-CoV-2 (Wuhan Hu-1) compared to patients treated with ustekinumab. Our study also revealed higher anti-S-IgG levels in IBD patients treated with vedolizumab compared to anti-TNF therapy, though this difference was not statistically significant, possibly due to the limited number of participants in the subgroups. In the cohort of IBD patients and controls with COVID-19, no significant differences were observed in anti-S-IgG levels. However, it is important to note the limitation of the small sample size. Nevertheless, it appears that the overall levels of anti-RBD IgG antibodies alone do not predict the risk of breakthrough infections in IBD patients [26]. Functional neutralizing antibodies, used in the surrogate neutralization test we employed in our study, correlate with the risk of breakthrough infection and serve as a good indicator of humoral protection following SARS-CoV-2 vaccination [27]. Most of the aforementioned recent studies conducted after the third vaccination only assessed anti-S-IgG levels without considering the analysis of neutralization capacity. An exception is the CLARITY-IBD study. In this study, the neutralization capacity of antibodies was measured, and it was found that antibodies from patients treated with infliximab or a combination therapy with thiopurines exhibited significantly lower neutralizing capacities against wild-type SARS-CoV-2 (Wuhan Hu-1) and the Omicron subvariants BA.1 and BA.4/5 after three doses compared to antibodies from patients receiving vedolizumab [26,27]. However, the Clarity study did not investigate healthy control individuals. Regarding neutralizing antibodies against the wild-type virus, our study demonstrated a significantly reduced sVNT inhibition capacity in the COVID-19-negative IBD patient cohort 60 to 120 days after the third SARS-CoV-2 mRNA vaccination compared to healthy control individuals. However, in the subgroup analysis of IBD patients based on the presence of immunosuppression, no significant differences were found. This could again be attributed to small case numbers in the subgroup analyses. Additionally, compared to the CLARITY study, our study examined a longer interval from the third vaccination (60–120 days after the third vaccination, compared to 14–70 days after the third vaccination in CLARITY-IBD). The extended time interval in our study might explain the similar neutralization capacity observed in the subgroup analyses. No significant differences were observed in the SARS-CoV-2-positive cohort (sVNT IBD vs. controls). To the best of our knowledge, our prospective IBD study is the first to investigate follow-up of up to 4 months after the third SARS-CoV-2 vaccination. 

Edger et al. demonstrated in a large population-based study that individuals with inflammatory bowel disease (IBD) do not have an elevated risk of SARS-CoV-2 infection compared to healthy control subjects [28]. The severity of COVID-19 in IBD patients appears to be influenced by multifactorial factors [29]. Particularly, patients with increased IBD activity are at a higher risk of experiencing severe or critical COVID-19 illness, which may lead to hospitalization, oxygen requirement, invasive ventilation and death [30]. Another significant influencing factor is the ongoing pharmacological treatment for IBD. Therapies with systemic steroids, sulfasalazine, 5-aminosalicylic acid and thiopurines have been identified as risk factors for a severe course of COVID-19. In contrast, immunosuppressive therapies involving ustekinumab, vedolizumab and anti-TNF agents do not appear to be associated with severe COVID-19 disease progression [29,31,32,33]. Patients receiving treatment with anti-TNF seem to have an increased risk of penetrating infections, but COVID-19 typical symptoms and severe courses of disease are rare [26]. A possible explanation could be the inhibition of severe systemic inflammatory reactions in the context of COVID-19 by anti-TNF drugs. Furthermore, studies suggest that anti-TNF treatment is associated with a lower risk of COVID-19-related hospitalization and death [34,35,36]. However, our study found that IBD patients on anti-TNF therapy who had a breakthrough infection after a third vaccination had significantly higher anti-S-IgG levels compared to uninfected patients on anti-TNF therapy. These results confirm previously published data showing higher antibody levels after penetrating infections [27]. Whether this effectively protects against further infections needs to be explored in prospective studies involving immunosuppressed IBD patients. Our data corroborate previous studies that were unable to demonstrate significant differences in terms of the humoral response following COVID-19 vaccination based on the utilized anti-TNF agent (adalimumab, infliximab) [37].

Limitations: While most neutralizing antibodies target epitopes within the RBD, thus blocking viral attachment to its cellular receptor ACE2, other epitopes on the spike protein of SARS-CoV-2 have been delineated that mediate neutralization, such as the N-terminal domain [38,39,40,41]. As the tests used in our study are limited to the RBD as the diagnostic antigen, a small fraction of neutralizing antibodies in patient sera may have been potentially overlooked. Another limitation in our study was the examination of the humoral response after COVID-19 vaccination against the Wuhan ancestral strain. Further studies on current variants are necessary. Due to these limitations, no general treatment recommendations for IBD patients can be derived. In particular, IBD patients under anti-TNF therapy appear to have no increased risk for critical COVID-19 courses, despite their slightly reduced humoral immunity. In our study, the difference in humoral immunity, especially in sVNT, was minimal and at a high level.

## 5. Conclusions

In our study, we assessed anti-S-IgG levels and sVNT inhibition levels in COVID-19-negative and -positive IBD patients 60 to 120 days after the third SARS-CoV-2 vaccination. Immunosuppressed IBD patients without a previous SARS-CoV-2 infection exhibited a significantly reduced humoral immunity (anti-S-IgG, sVNT) compared to healthy control individuals, particularly patients under anti-TNF therapy. Breakthrough infections led to significantly higher antibody levels in IBD patients under anti-TNF therapy compared to noninfected patients under anti-TNF therapy. In the group of SARS-CoV-2-positive IBD patients and controls, no significant differences in humoral immunity were observed. However, it is important to highlight the limitation of the small sample size. Whether this effectively protects against further infections needs to be explored in prospective studies involving immunosuppressed IBD patients.

## Figures and Tables

**Figure 1 vaccines-11-01411-f001:**
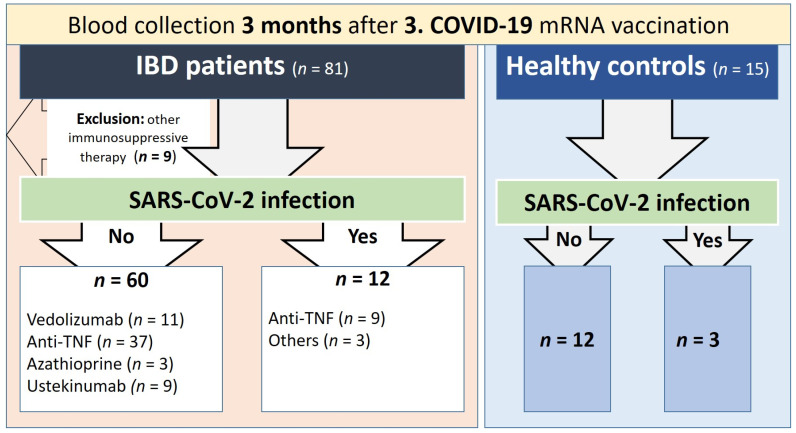
Study flow chart; IBD = inflammatory bowel disease.

**Figure 2 vaccines-11-01411-f002:**
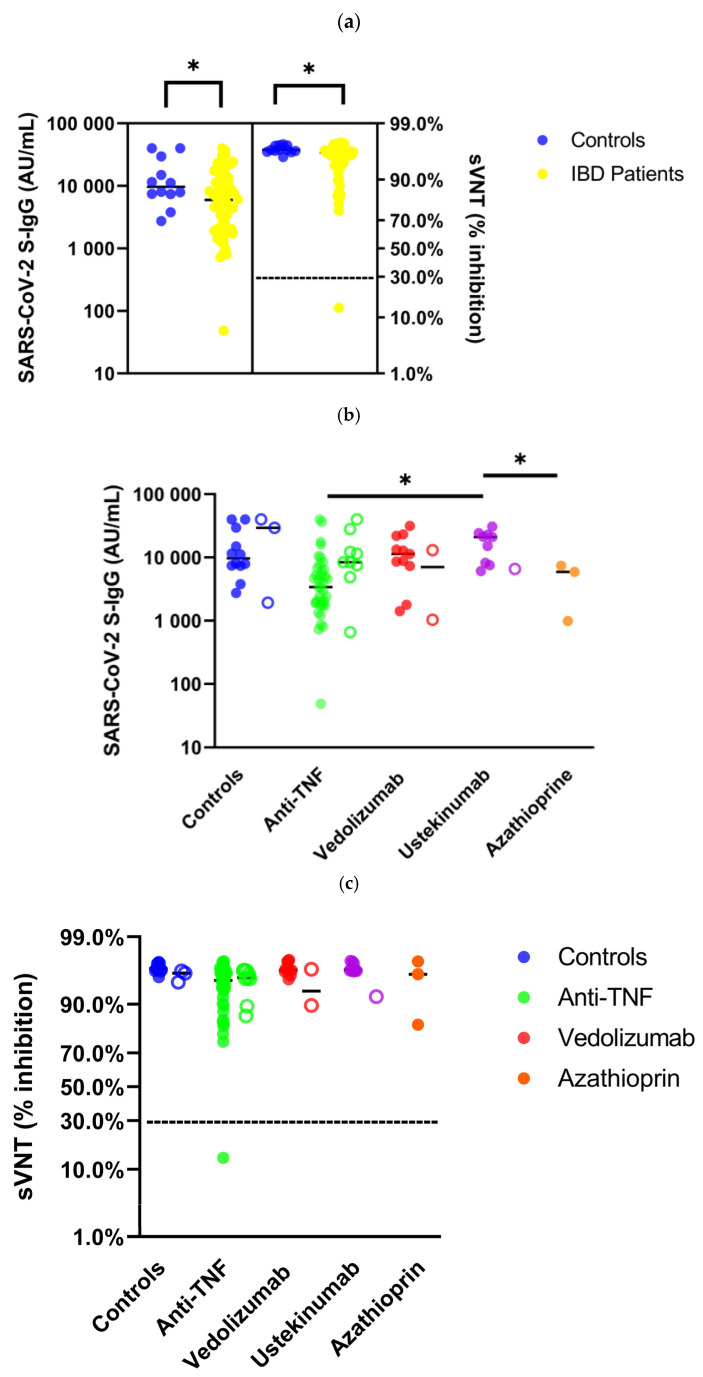
Humoral immunity in IBD patients without (•) and with (**◦**) previous SARS-CoV-2 infection 3 months after the third vaccination. Comparison of SARS-CoV-2-S IgG (AU/mL) and sVNT (% inhibition) of healthy controls with immunosuppressed IBD patients (**a**). Subgroup analysis of SARS-CoV-2-S IgG (**b**) and sVNT values (**c**) of IBD patients in relation to existing immunosuppressive medication and healthy controls. IBD, inflammatory bowel disease; * *p* < 0.05, cut-off value VNT 30% inhibition.

**Figure 3 vaccines-11-01411-f003:**
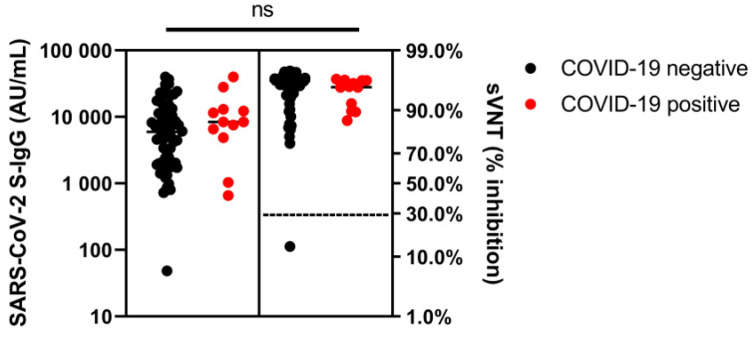
Plot of humoral reactivity (SARS-CoV-2 S-IgG levels, sVNT values) in IBD patients without SARS-CoV-2 infection and with previous SARS-CoV-2 infection 3 months after the third vaccination. ns, not significant.

**Table 1 vaccines-11-01411-t001:** Baseline characteristics of the study population. IBD, inflammatory bowel disease; IQR, interquartile range; BMI, body mass index; CDAI score, Crohn’s disease activity index. *p*-value ^1^: Mann–Whitney U test (Wilcoxon); *p*-value ^2^: Kruskal–Wallis test. Continuous variables were analyzed using the *t*-test if they followed a normal distribution and the Mann–Whitney U test (Wilcoxon) if the distribution was non-normal. The Kruskal–Wallis test was employed for comparisons involving more than two groups. For subgroup comparisons, the Bonferroni correction post hoc test was utilized in cases of equal variance (tested by Levene’s test), while the Games–Howell test was applied in cases of differing variance.

		No Previous SARS-CoV-2 Infection								Previous SARS-CoV-2 Infection					
		Controls (*n =* 12)	IBD (*n* = 60)	*p*-Value ^1^	Vedolizumab (*n* = 11)	Anti-TNF (*n* = 37)	Azathioprine (*n* = 3)	Ustekinumab *(n* = 9)	*p*-Value ^2^	Controls (*n* = 3)	IBD (*n* = 12)	*p*-Value ^1^	Anti-TNF (*n* = 9)	Others *(n* = 3)	*p*-Value ^2^
Patient characteristics	Age, years median (IQR)	46 (33–57)	50 (35–57)	0.678	50 (40–66)	52 (34–57)	48	48 (37–56)	0.780	36	39 (29–47)	0.734	38 (26–44)	46	0.104
	Sex, male (%)	42	37 (62)	0.219	10 (91)	22 (60)	0 (0)	5 (56)	0.027	1 (33)	7 (58)	0.003	4 (44)	3 (100)	0.108
	BMI	22 (21–25)	24 (23–27)	0.125	26 (23–28)	24 (23–27)	23	24 (23–26)	0.454	21	23 (22–25)	0.101	23 (21–25)	25	0.314
IBD	Crohn’s disease (%)	0 (0)	37 (62)		4	24	2 (67)	7 (78)	0.255	0 (0)	10 (83)		8 (89)	2 (67)	0.418
	CDAI score, median (IQR)	0 (0)	0 (0–0)		34 (0–205)	0 (0–0)	0 (0–0)	0 (0–0)	0.207	0 (0)	0 (0–30)		0 (0–0)	59	0.439
	CDAI score >/=150 (%)	0 (0)	2 (5)		1 (9)	0 (0)	0 (0)	1 (11)	0.141	0 (0)	0 (0)		0 (0)	0 (0)	1.000
	Ulcerative colitis (%)	0 (0)	23 (38)		7	13	1 (33)	2 (22)	0.207	0 (0)	2 (17)		1 (11)	1 (33)	0.418
	Mayo score, median (IQR)	0 (0)	0 (0–4)		3 (0–4)	0 (0–2.5)	0 (0–0)	2,5	0.282	0 (0)	0 (0–0)		0 (0–0)	0	1.000
	Mayo score >/=1 (%)	0 (0)	11 (48)		5 (46)	4 (31)	0 (0)	2 (100)	0.107	0 (0)	0 (0)		0 (0)	0 (0)	1.000
Medication	Prednisolone p.o. (%)	0 (0)	4 (6)		1 (9)	3 (8)	0 (0)	0 (0)	0.794	0 (0)	1 (8)		0 (0)	1 (33)	0.082
	Budesonide p.o. (%)	0 (0)	1 (1)		1 (9)	0 (0)	0 (0)	0 (0)	0.218	0 (0)	1 (8)		1 (11)	0 (0)	0.588
	Budesonide supp. (%)	0 (0)	3 (4)		1 (9)	2 (5)	0 (0)	0 (0)	0.806	0 (0)	1 (8)		0 (0)	0 (0)	1.000
	Mesalazine p.o. (%)	0 (0)	25 (34)		4 (36)	13 (35)	2 (67)	6 (67)	0.289	0 (0)	4 (33)		2 (22)	2 (67)	0.188
	Mesalazine supp. (%)	0 (0)	5 (7)		1 (9)	2 (5)	1 (33)	1 (11)	0.413	0 (0)	0 (0)		0 (0)	0 (0)	1.000
Pre-existing conditions	Cardiovascular disease	0 (0)	12 (17)		4 (36)	8 (22)	0 (0)	0 (0)	0.186	0 (0)	0 (0)		0 (0)	0 (0)	1.000
	Respiratory disease (%)	0 (0)	3 (4)		2 (18)	1 (3)	0 (0)	0 (0)	0.173	0 (0)	2 (17)		1 (11)	1 (33)	0.418
	Kidney insufficiency (%)	0 (0)	2 (3)		1 (9)	1 (3)	0 (0)	0 (0)	0.678	0 (0)	1 (8)		1 (11)	0 (0)	0.588
	Metastatic neoplasm (%)	0 (0)	4 (6)		2 (18)	2 (5)	0 (0)	0 (0)	0.363	0 (0)	1 (8)		0 (0)	1 (33)	0.082
	Diabetes (%)	0 (0)	1 (2)		1 (9)	0 (0)	0 (0)	0 (0)	0.218	0 (0)	0 (0)		0 (0)	0 (0)	1.000
	Death (%)	0 (0)	0 (0)		0 (0)	0 (0)	0 (0)	0 (0)	1	0 (0)	0 (0)		0 (0)	0 (0)	1.000

**Table 2 vaccines-11-01411-t002:** Humoral immune response in IBD patients and healthy controls, stratified by previous SARS-CoV-2 infection. The table presents SARS-CoV-2 S-IgG levels (AU/mL) and sVNT (% inhibition) values of IBD patients and healthy controls three months after the third vaccination. The IBD patient cohort was segmented based on their specific immunosuppressive therapies, including vedolizumab, anti-TNF agents (adalimumab, infliximab), azathioprine, ustekinumab and other therapies (azathioprine, n = 2; ustekinumab, n = 1). IBD, inflammatory bowel disease; IQR, interquartile range. *p*-value ^1^: Mann–Whitney U test (Wilcoxon); *p*-value ^2^: Kruskal–Wallis.

COVID-19-Negative *	Controls (*n* = 12)	IBD (*n* = 60)	*p*-Value ^1^	Vedolizumab (*n* = 11)	Anti-TNF (*n* = 37)	Azathioprine (*n* = 3)	Ustekinumab *(n* = 9)	*p*-Value ^2^
SARS-CoV-2 S-IgG (AU/mL), median (IQR)	9627 (7420–25,950)	5956 (1996–12,461)	0.034	11,388 (7319–22,027)	3410 (1843–6617)	5903 (989–5903)	20,928 (7882–23,558)	<0.001
Seroconversion rate S-IgG (%)	100	98	1.000	100	97	100	100	0.889
sVNT (% inhibition), median (IQR)	97 (96–97)	96 (94–97)	0.012	96 (96–97)	95 (90–96)	96 (83–96)	97 (96–97)	<0.001
Seroconversion rate sVNT (%)	100	98	1.000	100	97	100	100	0.889
**COVID-19-Positive ***	**Controls** **(*n* = 3)**	**IBD** **(*n* = 12)**	***p*-Value ^1^**	**Others (n = 3)**	**Anti-TNF** **(*n* = 9)**			***p*-Value ^2^**
SARS-CoV-2 S-IgG (AU/mL), median (IQR)	29,365 (1934–29,365)	8409 (5314–12,846)	0.365	6577 (1032–6577)	8434 (6227–20,155)			0.518
Seroconversion rate S-IgG (%)	100	100	1.000	100	100			1.000
sVNT (% inhibition), median (IQR)	96 (95–96)	95 (90–96)	0.840	92 (90–92)	95 (92–96)			0.926
Seroconversion rate sVNT (%)	100	100	1.000	100	100			1.000

***** COVID-19 negative and COVID-19 positive collective.

## Data Availability

The data are not publicly available due to privacy/ ethical restrictions.

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
