# Peer review of "Humoral Immunity in Immunosuppressed IBD Patients after the Third SARS-CoV-2 Vaccination: A Comparison with Healthy Control Subjects"

_vaccines, 2023, doi:10.3390/vaccines11091411_

Round 1

Reviewer 1 Report

The study of Vollenberg et al. used data from anti-TNF (infliximab, adalimumab), anti-α4ß7 integrin (vedolizumab), ustekinumab and azathioprine-treated ID patients with Covid vaccination and tried to investigate the pattern of humoral response post covid vaccination in immunocompromised patients. The study is well-designed and statistically well taken care of. However, I found a few points that should be taken care of. 

  1. In the abstract, they gave that as anti-TNF, they used infliximab and adalimumab, but in the text, it needs to be clarified which of these two is used since different mAbs might pose different effects. 
  2. It is not evident that the ustekinumab and Azathioprine are targeted against witch ligand. Authors are suggested to include this part also. 
  3. In lines 170-171, it is said that “The RBD-ACE2 binding inhibition, S-IgG levels, and seroconversion rates VNT, S- 170 IgG) of IBD patients and healthy controls after SARS-CoV-2 infection did not differ significantly,” but in abstract, they wrote that “Patients receiving anti-TNF therapy showed significantly reduced anti-S- 25 IgG levels after booster vaccination compared to those receiving ustekinumab and azathioprine 26 (p=0.030, p=0.031)”. The authors need to clear this discrepancy.
  4. What is the takeaway conclusion of the authors? Do they want to suggest a preference to be given to anti-TNF treatment other treatment with Azathioprine, vedolizumab, and ustekinumab of IBD patients where the chances of Covid infections are also present?

Author Response

We thank the reviewer for his good advice to improve the publication. Please see the attachment.

Reviewer 2 Report

There are many studies already published on the said topic.

For detailed comments, pls see attached file.

There are many studies already published on the said topic.

For detailed comments, pls see attached file.

Reviewer 3 Report

In this manuscript, the authors aimed to investigate the humoral response in immunosuppressed patients with IBD after receiving the third mRNA vaccination against SARS-CoV-2. The study found that IBD patients, particularly those on anti-TNF therapy, had a reduced humoral immune response compared to healthy individuals. Additionally, patients with penetrating infections had significantly higher antibody levels compared to uninfected patients, which has it certain value. However, there are some problems, which must be solved before it is considered for publication.

1. Introduction

1) You are supposed to improve the readability and flow of the sentences by organizing the information more clearly. Consider breaking down complex sentences into shorter, simpler ones.

2) Clarify the purpose of the study in the introduction, highlighting the specific aim of investigating the humoral response in immunosuppressed IBD patients after COVID-19 vaccination.

2. Methods

1) On page 2, line 71-74, the authors defined a positive COVID-19 nasopharyngeal swab PCR test as a previous COVID-19 infection. However, it is unclear whether the authors have excluded patients who are currently infected with COVID-19.

2) On page 2, line 77-79, it is mentioned that 13 patients after COVID-19 infection were included, however, in Figure 1, it states that 12 patients after COVID-19 infection were included. Please review and amend this discrepancy accordingly.

3) In Figure 1, I suggest that the authors could increase the font size and adjusting the background color of the text boxes in the image to improve clarity and visual appeal.

4) On page 3, line 114, please unify the format of p-value in this manuscript. The p-value in the text is not italicized, while the p-value in the tables is italicized.

5) Please provide more details about the methodology used in the study, including the specific vaccines administered and the timing of the vaccinations in relation to the patients' treatment.

3. Results

Please unify the font size and format of Table 1 and Table 2.

4. Discussion

The discussion in this section lacks a thorough exploration of the significance of the research findings, particularly concerning the protective effects in preventing future infections and severe diseases. It is recommended to provide a more comprehensive and detailed analysis in this section.

5. References

Please unify the format of the first reference. In addition, you should ensure that all references are correctly formatted again, including the proper use of punctuation, capitalization and consistent citation style.

You are supposed to improve the readability and flow of the sentences by organizing the information more clearly. Consider breaking down complex sentences into shorter, simpler ones.

Reviewer 4 Report

Vollenberg et al. investigate the humoral immunity of immunosuppressed IBD patients after a third SARS-CoV-2 vaccination. In their study they enrolled 72 IBD patients with and without previous SARS-CoV-2 infection and compared the RBD IgG levels by ELISA with two methods, indirect ELISA against RBD or competition ELISA using ACE2 receptor as a surrogate of neutralizing antibodies. The aim, methodology and results are clearly presented in the manuscript.

Major comments

The authors state that IBD patients have a significant reduction on humoral immunity in their study, which is supported by the statistical analysis. However, the quantitative difference is not that big, specially when measuring neutralizing activity: IBD patients median 96% [94-97%] vs. controls 97% [IQR 96-97%]. Given this minimal difference I would be more careful about the conclusions of a lower protection of IBD patients. The n of the study is very low, especially in healthy individuals and a high heterogeneity of immune response has been seen in other cohorts compared to the n=15 samples analyzed here.

Authors only studied humoral response against the Wuhan ancestral strain a which does not circulate since 2021. It will be good to see if there is more remarkable antibody difference against the current variants of concern, like the XBB.1.5 or the XBB.1.16 strain that is taking over worldwide.

Finally, authors only study the humoral immune response at one time point, which weakens the conclusion.

Round 2

Reviewer 2 Report

Accept

Accept

Reviewer 3 Report

I have carefully reviewed the authors revised submission and I think the revisions have significantly improved the clarity, organization, and overall quality of the manuscript. The changes the authors made have adequately addressed the comments and suggestions provided by the reviewers. Overall, congratulations on the significant progress made with the manuscript. 

Reviewer 4 Report

No further comments